# Associations of maximum and reactive strength indicators with force–velocity profiles obtained from squat jump and countermovement jump

**Takuya Nishioka**[1]*, **Junichi Okada**[2]

**1** Graduate School of Sport Sciences, Waseda University, Tokorozawa, Saitama, Japan, **2** Faculty of Sport Sciences, Waseda University, Tokorozawa, Saitama, Japan

* w-t.n.89@fuji.waseda.jp

**Data Availability Statement:** All relevant data are within the manuscript and its Supporting Information files.

## Abstract

Understanding the properties associated with the vertical force–velocity (F–v) profiles is important for maximizing jump performance. The purpose of this study was to evaluate the associations of maximum and reactive strength indicators with the F–v profiles obtained from squat jump (SJ) and countermovement jump (CMJ). On the first day, 20 resistance-trained men underwent measurements for half squat (HSQ) one-repetition maximum (1RM). On the second day, jump performances were measured to calculate the drop jump (DJ) reactive strength index (RSI) and the parameters of F–v profiles (theoretical maximum force [F0], velocity [V0], power [Pmax], and slope of the linear F–v relationship [SFv]) obtained from SJ and CMJ. The DJ RSI was not significantly correlated with any parameter of the vertical F–v profiles, whereas the relative HSQ 1RM was significantly correlated with the SJ F0 ($r = 0.508$, $p = 0.022$), CMJ F0 ($r = 0.499$, $p = 0.025$), SJ SFv ($r = -0.457$, $p = 0.043$), and CMJ Pmax ($r = 0.493$, $p = 0.027$). These results suggest that maximum strength is a more important indicator than reactive strength in improving vertical F–v profiles. Furthermore, the importance of maximum strength may vary depending on whether the practitioner wants to maximize the performance of SJ or CMJ.

## Introduction

Jump performance (i.e., jump height) may be significantly related to athletic performance, such as sprinting [1] and change of direction [2]. Therefore, maximization of jump performance will be important for many athletes. The two commonly used tests to monitor jump performance in strength and conditioning as well as in research are the squat jump (SJ) and the countermovement jump (CMJ) [3–5]. The SJ is supposed to provide an assessment of the capability to rapidly develop force solely during a purely concentric movement, whereas the CMJ is supposed to provide an assessment of the capability to quickly produce force in stretch-shortening cycle (SSC) movements [4]. Thus, since the performance of SJ and CMJ is

**Funding:** The authors received no specific funding for this work.

**Competing interests:** The authors have declared that no competing interests exist.

expected to reflect different properties, the development of appropriate training strategies for both SJ and CMJ is necessary to maximize the performance of both movements.

The vertical force–velocity (F–v) profile, which provides information on the theoretical maximum force (F0), velocity (V0), power (Pmax), and slope of the linear F–v relationship (SFv), has recently been employed to evaluate jump performances of SJ and CMJ [6, 7]. F0 represents the maximal concentric force output (per body mass unit) that an athlete's lower limbs can theoretically produce during a ballistic push-off, and V0 represents the maximal extension velocity of the athlete's lower limbs during a ballistic push-off [7]. Both parameters are extrapolated from the linear F–v relationship in loaded jump squats, but not from peak values during a single jump [7]. Pmax, computed as Pmax = F0·V0/4, represents the maximal power output capability of the athlete's lower limb neuromuscular system (per body mass unit) in concentric and ballistic extension motions, and SFv is an index of the athlete's individual balance between force and velocity capabilities, in which there is an optimal balance for a given individual to maximize the jump performance [7]. Samozino et al. [8] suggested that high ballistic performance (i.e., jumping) is determined by the maximization of the Pmax and optimization of the SFv. In addition, Jiménez-Reyes et al. [9] reported that optimizing SFv without increasing the Pmax led to clearly beneficial changes in jump performance. Thus, since the vertical F–v profile is thought to play a significant role in maximizing jump performance, training based on it has been widely practiced [9–11].

Furthermore, maximum and reactive strengths are used as typical strength performance indicators that improve jump performance [12, 13]. In general, maximum strength is evaluated by the one-repetition maximum (1RM) of a resistance exercise (e.g., half squat [HSQ]) [14], and reactive strength is often assessed through the reactive strength index (RSI) during a drop jump (DJ) [15]. Maximum strength is important to reduce the risk of injury [16] and improve athletic performance [17]. Reactive strength has been reported to be an indicator of the ability to utilize fast SSC and is associated with jumping [18], sprinting [19], and change of direction performance [20]. Therefore, training to increase maximum and reactive strengths is extensively practiced for strength and conditioning. Given the aforementioned importance of F–v profiling, understanding the properties associated with the vertical F–v profile is important in maximizing the jump performance. However, how the strength indicators relate to the parameters of the F–v profiles of SJ and CMJ remains unclear. From a practical perspective, this information can help practitioners design training programs to maximize the performance of SJ and CMJ.

Therefore, the purpose of this study was to evaluate the associations of maximum and reactive strength indicators with the parameters of F–v profiles obtained from SJ and CMJ. We hypothesized that the maximum and reactive strength indicators would be positively associated with the vertical F–v profiles (i.e., F0, V0, and Pmax). In the present study, we analyzed data from a previous study [21] for a completely different purpose.

## Materials and methods

### Experimental design

This cross-sectional study was conducted across 2 days. On the first day, the participants were familiarized with the experimental trials used on the second day, and preliminary measurements were taken to determine the optimum drop height and HSQ 1RM for each participant. On the second day, jump performance (0% and 40% 1RM SJ, 0% and 40% 1RM CMJ, and DJ from the optimum drop height) was evaluated to calculate DJ RSI and the parameters of the F–v profiles (F0, V0, Pmax, and SFv) obtained from SJ and CMJ. The test days were separated by 72–144 h.

## Participants

This study was conducted in Japan from December 2020 to April 2021. Sample size calculations (G*Power software, version 3.1.9.4, Dusseldorf, Germany) were performed to determine the sample size needed to analyze the correlation between strength performance and F–v profile variables (to detect a correlation of 0.6 between variables) [22]. With a significance value of 0.05 and a power of 0.80, a sample size of 17 participants was calculated for this analysis. Hence, 20 resistance-trained men between the ages of 20 to 25 (baseball players: $n = 7$; American football players: $n = 4$; basketball players: $n = 3$; other sports players: $n = 6$) (age: 22.4 ± 1.5 years, height: 172.2 ± 5.0 cm, body mass: 71.3 ± 7.4 kg, HSQ 1RM: 142.5 ± 28.4 kg) were recruited to participate in this study. The included participants had an average sports training background of 11.6 ± 3.7 years and 4.3 ± 2.4 years of resistance training experience; they were free from musculoskeletal pain or injury that can compromise testing. To minimize confounding factors, instructions related to sleep and diet were provided to the participants before the experiment. On the night preceding each test session, the participants were asked to maintain their usual sleeping habits, with a minimum sleep of 7 h. During the investigation period, the participants were requested to avoid the intake of any known stimulants (e.g., caffeine) or depressants (e.g., alcohol) that can enhance or compromise their wakefulness. Moreover, the participants were instructed to maintain their habitual physical activity level and avoid strenuous activity the day before and throughout the study. After explaining the purpose, procedures, risks, and benefits of the study to the potential participants, written informed consent was obtained before participation. This study was approved by the Ethics Review Committee on Human Research of Waseda University (Approval number: 2020–267). All procedures in this study were conducted according to the Declaration of Helsinki.

## Familiarization and preliminary measurements

During the preliminary session, the participants were familiarized with the SJ (unloaded and loaded); CMJ (unloaded and loaded); DJ from 20, 40, and 60 cm heights; and jump squat with 40% estimated 1RM. Furthermore, the participants performed three repetitions of DJ from 20, 40, and 60 cm with adequate rest (60–90 s) to determine the individual optimum drop height for the main trials. The optimum drop height was defined as the height with the greatest RSI (= DJ height/ground contact time). This approach was selected based on the available literature, highlighting the importance of identifying the individualized optimum drop height to maximize neuromuscular adaptations [23, 24]. After 3 min of rest, the HSQ 1RM of each participant was measured. The maximum strength of the lower body was measured using a knee angle of 90° since the HSQ strength is strongly correlated with jump performance [13]. HSQ 1RM testing was conducted by letting the participants complete a series of warm-up sets (five repetitions at 30% estimated 1RM, three repetitions at 50% estimated 1RM, two repetitions at 70% estimated 1RM, and one repetition at 90% estimated 1RM), each separated by 3 min of recovery. Then, a series of maximal lift attempts was performed until 1RM was obtained. The participants performed the downward movement of the HSQ exercise until the lowest position was reached, which was determined by the beep sound that was activated when the photocell beam of the timing gate (Brower Timing Systems, Draper, Utah, USA) was interrupted by the posterior portion of the thigh at a knee angle of 90°. The knee angle at the lowest position of movement of the HSQ was filmed from the sagittal plane (a distance of 5 m) using a smartphone video camera placed on a tripod at 240 Hz and a height of 40 cm (iPhone 7, Apple Inc., Cupertino, CA, USA). A two-dimensional motion analysis was then conducted on the data obtained using the Kinovea video analysis software (v. 0.8.15). The knee angle was calculated by digitizing the reflective markers attached to the greater trochanter, lateral epicondyle of the

femur, and lateral malleolus. The line connecting the greater trochanter to the lateral epicondyle of the femur, as well as the line connecting the lateral epicondyle of the femur to the lateral malleolus created an angle that was defined as the knee angle. Trials in which the participants did not reach a knee angle of less than 90° of flexion were excluded from the analysis.

### Performance measurement session

First, 72–144 h after the preliminary session, the participants performed a standardized warm-up that included 3 min of jogging, 10 min of dynamic stretching, and two repetitions each of SJ and CMJ at sub-maximal effort (at approximately 80% of maximal effort), followed by two repetitions each of SJ and CMJ with 0% 1RM, 20 kg, 30% 1RM, and 40% 1RM with adequate (60–90 s) rest. After completing this standardized warm-up, the participants performed three repetitions each of SJ and CMJ at 0% 1RM with 90 s of rest, three repetitions each of SJ and CMJ at 40% 1RM with 2 minutes of rest, and three repetitions of DJ with 90 seconds of rest. The order of these jump performance measurements was randomized. All jump performance measurements were performed at the same time of day between 14:30 and 19:00 h to avoid diurnal variations. Ad libitum drinking was permitted to prevent dehydration, and all testing sessions were performed under the same environmental conditions (18°C–22°C and 20%–60% humidity).

### Measurements of SJ, CMJ, and DJ performances

To perform the SJ and CMJ, the participants held a 0.1-kg plastic bar for 0% 1RM and a 20-kg barbell loaded with the appropriate weight plates for 40% 1RM. All loaded jump trials were performed with free weights. During the SJ, the participants were instructed to lower into an HSQ position at the desired knee angle, indicated by a beep sound, and hold this position for 2 s. As required, they then jumped as high as possible without performing a previous countermovement. We checked the waveform data obtained from the force plate during the trial to confirm that no previous countermovement was used. For the CMJ, the participants were instructed to perform the downward movement as deep as the SJ, perform a countermovement as fast as possible, and jump as high as possible. For the DJ, the participants were asked to step off a wooden box at a set height without lifting their center of gravity and land on the force plate with both legs. In addition, they were instructed to rebound and immediately jump as high as possible after contact while minimizing ground contact time. Their hands were kept akimbo throughout the jump, while a straight body position during landing and take-off was encouraged. The participants were also required to land back on the force plate.

### Measurement equipment and data analyses

All jumps (SJ, CMJ, and DJ) were performed on a single force platform (0625, ACP, Accu-Power; AMTI, Watertown, MA, USA) that sampled the vertical ground reaction force (GRF) at a frequency of 1000 Hz using an analog-to-digital converter (EIRBZ22002369; CONTEC Co Ltd, Osaka, Japan). Then, data were recorded on a personal computer. Signals from the force plate were filtered by a 50-Hz low-pass, zero-phase-lag finite impulse response filter. Before each SJ and CMJ, the participants were weighed for 3 s with the external load laid on their shoulders to determine the total system weight (the sum of the body weight and the external weight). The start of the jump was defined as the time point 30 ms before the vertical GRF exceeded the threshold (the total system weight ± 5 $SD$) [25]. For each jump, the center of mass (COM) velocity of the system was calculated using the trapezoid rule [26], whereas the net GRF was calculated as the amount of force exceeding the system weight divided by the system mass to determine acceleration. Acceleration was numerically integrated to provide the

instantaneous COM velocity, which was, in turn, numerically integrated to provide instantaneous COM displacement. The beginning of the concentric phase was when the vertical COM velocity exceeded 0 m/s after starting the jump. Take-off was identified as the moment when the vertical GRF decreased to <20 N after the start of the concentric phase [27]. The contact and take-off of the DJ were defined as the time points, at which the vertical GRF exceeded or decreased below the threshold (i.e., 20 N), respectively. The DJ height was calculated through the flight time [28]. To obtain the F–v profiling of SJ and CMJ, the mean vertical force developed by the lower limbs during push-off (i.e., the CON phase) after which the corresponding mean COM vertical velocity was determined using equations validated by Samozino et al. [28] and Jiménez-Reyes et al. [6]. The total system weight, push-off distance, and jump height data substituted for Samozino's equations [28] were derived from the vertical GRF [26]. The push-off distance was identified as the distance of the vertical COM displacement from the start to the end of the concentric phase. The jump height was calculated after performing the take-off-velocity procedure [26]. Owing to the fact that the force–velocity relationship of vertical jumps is linear [7, 8], the force and velocity data obtained under two different loads (0% and 40% 1RM) were modeled using a least-squares linear regression model to determine the F–v profile, $F(V) = F0 - aV$, where F0 denotes the theoretical maximum force (i.e., the force-intercept), and V0 denotes the theoretical maximum velocity (i.e., the velocity-intercept) corresponding to the slope of the linear F–v relationship (SFv = −F0/V0) [8, 29]. The average push-off distance during the 0% and 40% 1RM was used in the analyses. The maximum power (Pmax) was calculated as Pmax = F0·V0/4. This two-point method was employed to minimize fatigue during performance testing based on distant loads as validated by García-Ramos et al. [30] as a quick and less tiring procedure for testing the F–v profile. All force, power, and maximum strength (i.e., HSQ 1RM) values were normalized to the participant's body mass. The maximum value of each variable was used in the analyses. In addition, the eccentric utilization ratio (EUR = CMJ variable/SJ variable) was calculated for F0, V0, and Pmax [4].

## Statistical analyses

Normality was evaluated using the Shapiro–Wilk test. The intra-trial reliability of each mechanical variable was calculated using the two maximal values from the testing session. We calculated the intraclass correlation coefficient (ICC; two-way mixed effects, absolute agreement, and single rater/measurement) [31] and the coefficient of variation (CV). The correlation strength was interpreted as slight, fair, moderate, substantial, and almost perfect for ICCs of ≤0.20, 0.21–0.40, 0.41–0.60, 0.61–0.80, and ≥0.81, respectively [32], and CVs lower than 10% were interpreted as acceptable [33]. The relationships between the performance outcomes were calculated using Pearson's correlation tests ($r$) or Spearman's correlation tests ($\rho$) when normal distributions were not observed (0.0–0.1: no association; 0.1–0.4: weak association; 0.4–0.6: moderate association; 0.6–0.8: strong association; >0.8: very strong association) [34]. Statistical analyses were conducted using SPSS (IBM SPSS Statistics Version 27), with $p \leq 0.05$ indicating statistical significance.

## Results

The ICCs for the jump performance and F–v profile variables demonstrated almost perfect reliability, with all acceptable CVs (Table 1). The relative HSQ 1RM had a moderately significant correlation with the unloaded CMJ height, SJ F0, CMJ F0, SJ SFv, and CMJ Pmax, but not with the unloaded SJ height, SJ V0, SJ Pmax, CMJ V0, and CMJ SFv (Fig 1). The DJ RSI did not significantly correlate with any vertical F–v profile parameter (Fig 2). Furthermore, the relative HSQ 1RM was also moderately associated with the Pmax EUR, but not with the F0 EUR

**Table 1. Means, standard deviations, and reliability measures for jump performance and force–velocity (F–v) profile variables data.**

|  | mean | SD | ICC (95% CI) | CV (%) |
|---|---|---|---|---|
| Unloaded SJ height (m) | 0.35 | 0.04 | 0.918 (0.398–0.978) | 2.14 |
| Unloaded CMJ height (m) | 0.38 | 0.04 | 0.959 (0.398–0.990) | 1.22 |
| DJ RSI (m/s) | 1.46 | 0.29 | 0.920 (0.122–0.981) | 5.51 |
| SJ F0 (N/kg) | 36.56 | 5.67 | 0.844 (0.543–0.942) | 2.77 |
| SJ V0 (m/s) | 2.76 | 0.63 | 0.877 (0.717–0.949) | 5.20 |
| SJ Pmax (W/kg) | 24.49 | 2.80 | 0.899 (0.714–0.962) | 2.96 |
| SJ SFv (N·s/kg/m) | −14.23 | 4.90 | 0.861 (0.669–0.943) | 8.80 |
| CMJ F0 (N/kg) | 36.33 | 4.21 | 0.932 (0.803–0.974) | 2.20 |
| CMJ V0 (m/s) | 2.78 | 0.33 | 0.890 (0.744–0.955) | 2.86 |
| CMJ Pmax (W/kg) | 25.06 | 2.50 | 0.928 (0.714–0.976) | 2.07 |
| CMJ SFv (N·s/kg/m) | −13.35 | 2.95 | 0.925 (0.825–0.970) | 4.89 |
| F0 EUR | 1.01 | 0.14 | 0.862 (0.682–0.943) | 3.66 |
| V0 EUR | 1.05 | 0.23 | 0.883 (0.731–0.952) | 6.23 |
| Pmax EUR | 1.03 | 0.10 | 0.870 (0.703–0.946) | 3.12 |

ICC = intraclass correlation coefficient; CI = confidence interval; CV = coefficient of variation; SJ = squat jump; CMJ = countermovement jump; DJ = drop jump; RSI = reactive strength index; F0 = theoretical maximum force; V0 = theoretical maximum velocity; Pmax = theoretical maximum power; SFv = slope of the linear F–v relationship; EUR = eccentric utilization ratio.

and V0 EUR (Fig 3). Finally, the F–v profile obtained from the SJ had a moderately significant correlation with that obtained from the CMJ in F0 and Pmax, but not in V0 and SFv (Fig 4).

## Discussion

The purpose of the present study was to evaluate the association of maximum and reactive strength indicators with the parameters of the F–v profiles obtained from the SJ and CMJ.

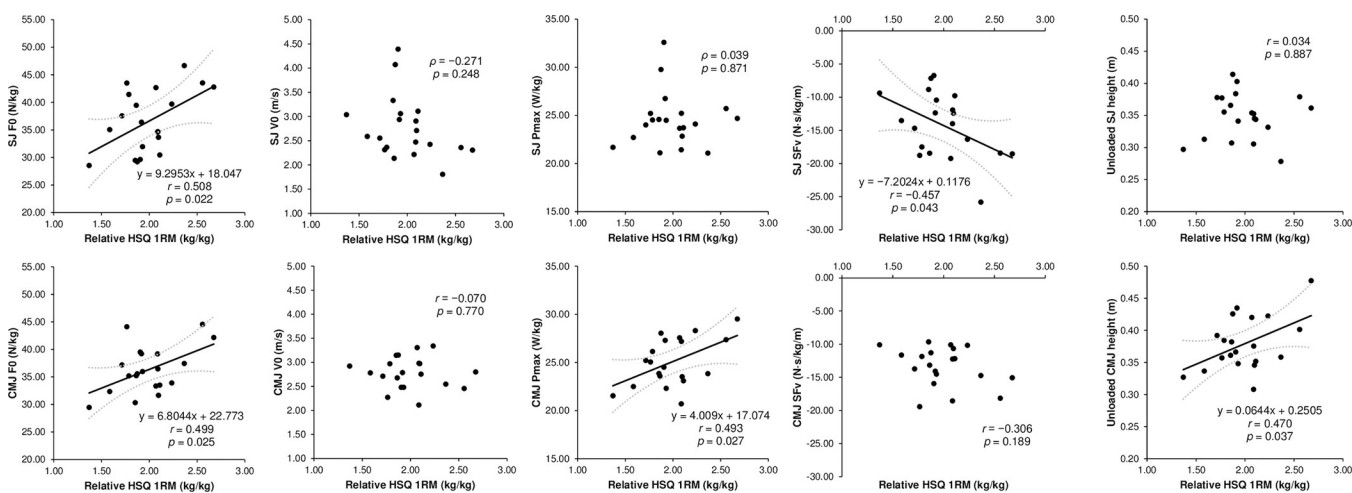

**Fig 1. Associations (line-of-best-fit ± 95% confidence intervals) of the maximum strength (relative half squat [HSQ] one-repetition maximum [RM]) with the parameters of the force–velocity (F–v) profiles obtained from the squat jump (SJ) and countermovement jump (CMJ) and with the unloaded SJ and CMJ height.** F0 = theoretical maximum force; V0 = theoretical maximum velocity; Pmax = theoretical maximum power; SFv = slope of the linear F–v relationship.

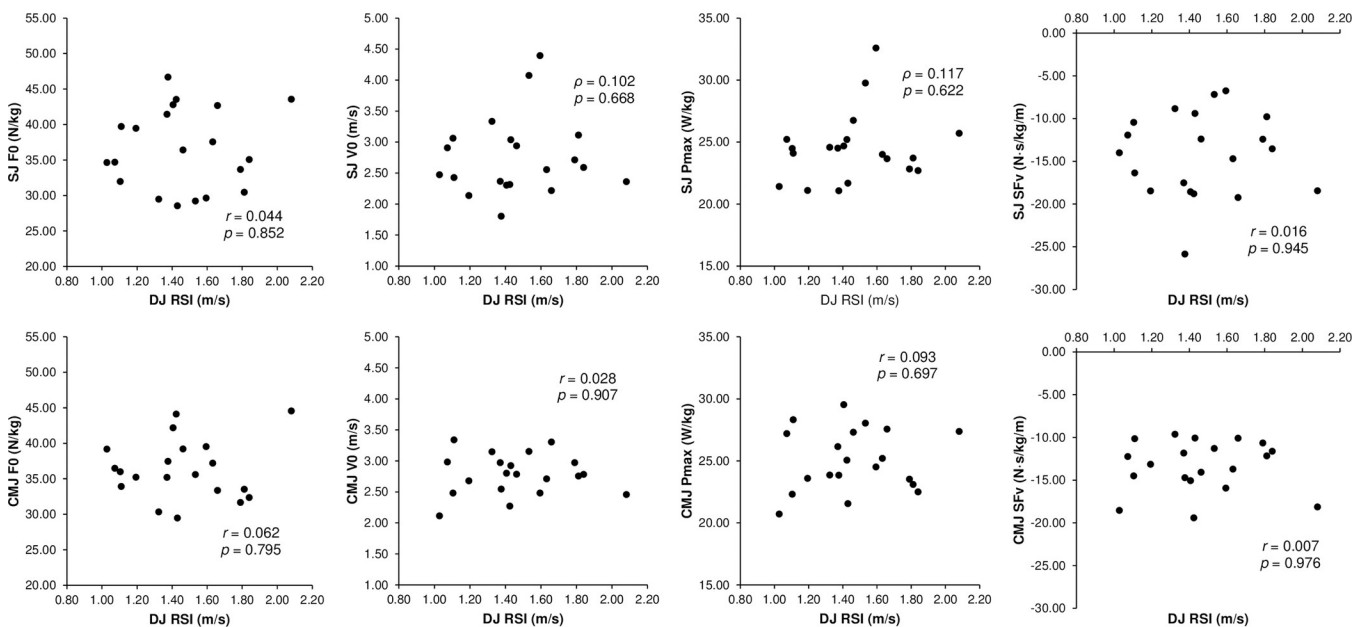

**Fig 2. Associations (line-of-best-fit ± 95% confidence intervals) of the reactive strength (drop jump [DJ] reactive strength index [RSI]) with the parameters of the force–velocity (F–v) profiles obtained from the squat jump (SJ) and countermovement jump (CMJ).** F0 = theoretical maximum force; V0 = theoretical maximum velocity; Pmax = theoretical maximum power; SFv = slope of the linear F–v relationship.

Therefore, although the maximum strength (i.e., the relative HSQ 1RM) was related to the theoretical maximum force (i.e., F0) obtained from the SJ and CMJ, the reactive strength (i.e., DJ RSI) was unrelated to any parameter of the vertical F–v profiles. We hypothesized that the maximum and reactive strength indicators would be positively associated with the vertical F–v profiles (i.e., F0, V0, and Pmax). These results confirmed our hypothesis partially for the maximum strength, but not for reactive strength. Accordingly, maximum strength can be a more important indicator than reactive strength in improving vertical F–v profiles. The relative HSQ 1RM was also associated with the SJ SFv and CMJ Pmax. Therefore, the importance of maximum strength is proposed to vary depending on whether the practitioner wants to improve the performance of SJ or CMJ.

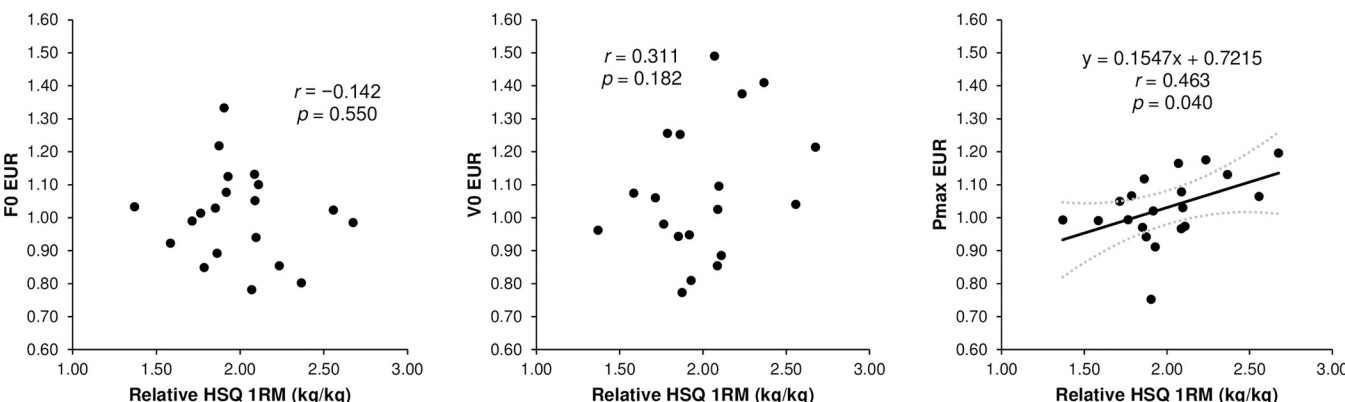

**Fig 3. Associations (line-of-best-fit ± 95% confidence intervals) of the maximum strength (relative half squat one-repetition maximum) with the eccentric utilization ratio (EUR) of the theoretical maximum force (F0), velocity (V0), and power (Pmax).**

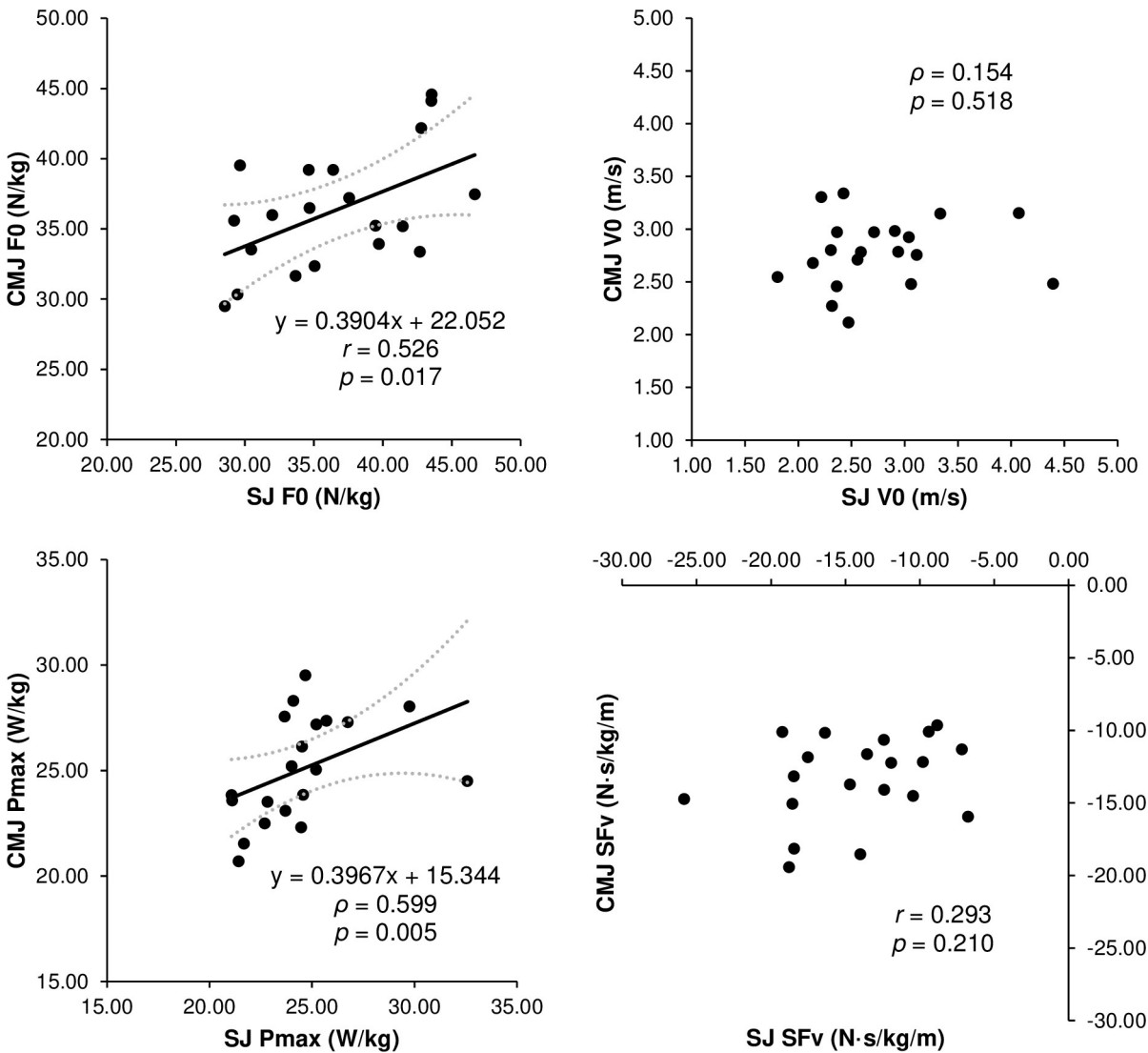

**Fig 4. Relationships (line-of-best-fit ± 95% confidence intervals) between the parameters of the force–velocity (F–v) profiles obtained from the SJ and CMJ.** F0 = theoretical maximum force; V0 = theoretical maximum velocity; Pmax = theoretical maximum power; SFv = slope of the linear F–v relationship.

This study demonstrates that the relative HSQ 1RM is related to the SJ F0 (Fig 1), consistent with a previous study by Rivière et al. [35]. The relative HSQ 1RM reflects the ability of force production at a mean propulsive velocity of 0.33 m/s [36], and F0 reflects the maximum concentric force output that the athlete's lower limbs can theoretically produce during ballistic push-off at null velocity [7, 8], indicating that relative HSQ 1RM and F0 are both indices that evaluate the ability of force outputs at low velocities. Therefore, they were considered related in the current study. Furthermore, Jiménez-Reyes et al. [9] reported that 9-week high-load resistance training (e.g., 80–90% 1RM back squats) enhanced the SJ F0. Such resistance training is also thought to improve maximum strength performance (e.g., 1RM) [37]. Considering these findings and the significant association of the relative HSQ 1RM with the CMJ F0 (Fig 1), high-load resistance training, such as enhancing the relative HSQ 1RM, is proposed to improve not only SJ F0 but also CMJ F0.

As mentioned above, although a significant positive correlation between the relative HSQ 1RM and F0, the correlation strength was not strong (Fig 1). This could be explained by the difference in the capabilities reflected by the relative HSQ 1RM and F0. For example, the relative HSQ 1RM strongly demonstrates the ability to pass the sticking point, which is the bottleneck of lifting performance; however, it does not reflect the ability to exert force after passing the sticking point [38]. Besides, F0 reflects the maximal concentric force output that the athlete's lower limbs can theoretically produce during the overall ballistic push-off [7]. In this regard, relative HSQ 1RM and F0 reflect different abilities. Furthermore, Rivière et al. [35] suggested that the difference between SJ F0 and the mean force developed during HSQ with 1RM load is affected by individual SJ V0. This finding indicates that even if relative HSQ 1RM is the same, F0 can vary depending on other F–v profile variables. Therefore, selection of performance indicators and assessment of the maximum forceoutput ability should be performed with caution.

The DJ RSI was not significantly associated with the vertical F–v profiles (Fig 2); however, Cronin & Hansen [18] reported that the reactive strength was significantly correlated with the CMJ height. In addition, Gehri et al. [12] suggested that DJ training is effective for improving SJ and CMJ height. Based on these findings, we hypothesized that the reactive strength would be associated with the vertical F–v profiles. However, this hypothesis could not be proved owing to the differences in participant characteristics from previous studies. In the study by Cronin & Hansen et al. [18], the participants were professional rugby players, whereas in that by Gehri et al. [12], the participants did not participate in competitive sports (i.e., not resistance-trained). In the current study, we enrolled resistance-trained men with different backgrounds in sports training. Therefore, in most resistance-trained men, reactive strength might not be important for improving parameters of vertical F–v profiles as well as vertical jump performance.

As mentioned above, the DJ RSI was not significantly correlated with any parameter of the F–v profiles obtained from the SJ and CMJ, which can be attributed to the different biomechanics of each jump. The DJ uses the SSC [39], whereas SJ uses the concentric-only movement [4]. Regarding DJ and CMJ, Bobbert et al. [39] demonstrated that knee and ankle joint moments and power output showed larger values during DJ than during CMJ and that hip joint moments exhibited larger values during CMJ than during DJ. Furthermore, as DJ uses fast SSC [39] and CMJ uses slow SSC [5], the speed of SSC also differs between DJ and CMJ. Based on these findings, DJ RSI and the F–v profiles of SJ and CMJ should be interpreted as performance indicators reflecting different physical abilities. Furthermore, Marshall et al. [40] reported that countermovement DJ similar to CMJ was more effective than the bounce DJ similar to DJ in enhancing CMJ height. Therefore, DJ training, such as improving the RSI, would not be suitable for improving the F–v profiles of SJ and CMJ.

The relative HSQ 1RM was also associated with the CMJ Pmax, but not with the SJ Pmax, suggesting that relative HSQ 1RM can contribute more to the force output ability at moderate velocities in the CMJ than in the SJ. Movement characteristics of the HSQ exercise, CMJ, and SJ can explain this finding. Although CMJ is an exercise that uses slow SSC and HSQ, SJ does not use SSC [5, 41, 42]. This finding proposed that the HSQ exercise is more similar to CMJ than SJ in terms of SSC use. Furthermore, muscle slack is proposed to be eliminated in high-load (e.g., 1RM load) HSQ when using an external load and in CMJ when using countermovement [43]. Consequently, both HSQ with 1RM load and CMJ require the ability to exert force while muscle slack is eliminated. However, it is difficult to eradicate muscle slack using external load or countermovement in SJ with no external light load (i.e., at moderate velocities). Hence, the ability to eliminate muscle slack by co-contracting the lower limb muscles is required [43]. Therefore, based on the above, relative HSQ 1RM can be a performance

indicator more related to exerting force abilities at moderate velocities in CMJ than SJ. Furthermore, the relative HSQ 1RM was also positively correlated with the Pmax EUR (Fig 2). This finding indicates that the force output capacity during the HSQ contributes to the ability to utilize SSC at moderate velocities. In summary, for CMJ, maximum strength in the HSQ can positively affect the ability of force production and the utilization of SSC at moderate velocities (i.e., Pmax), which is a contributing factor in maximizing jump height [8]. As observed, the relative HSQ 1RM was positively correlated with the unloaded CMJ height (Fig 1). Additionally, the relative HSQ 1RM was not associated with the CMJ SFv (Fig 1), suggesting that maximum strength training will not affect the CMJ SFv. Therefore, for enhancing CMJ height, improving maximum strength may be necessary regardless of the athlete's F–v profile. In contrast, the relative HSQ 1RM was negatively associated with the SJ SFv (Fig 1). Accordingly, for increasing SJ height, practitioners should consider that increasing the maximum strength can change the F–v profile of SJ to force-dominant. If the athlete has a velocity-deficit (i.e., force-dominant) profile, changing to force dominance can negatively impact jump height [8].

In this study, we investigated the association of maximum and reactive strengths with the parameters of F–v profiles obtained from the SJ and CMJ, but the cause–effect relationships remain unclear. Since cross-sectional correlations and longitudinal cause–effect relationships are not always consistent [44], care must be taken in interpreting the results. Meanwhile, the F–v profile parameters associated with maximum strength differed between SJ and CMJ. Previous studies [9–11] investigated the effect of training on the F–v profile of SJ, but not that of CMJ. To maximize the performance of SJ and CMJ, it can be important to evaluate their F–v profiles [4, 43, 45]. Nevertheless, to the best of our knowledge, no studies have simultaneously and longitudinally evaluated changes in the F–v profiles of SJ and CMJ. Therefore, future studies are required to examine the impact of training interventions (e.g., high-load resistance training) on the parameters of F–v profiles obtained from SJ and CMJ. Furthermore, the F–v profile obtained from the SJ was not significantly correlated with that obtained from the CMJ for some parameters (i.e., the V0 and SFv) in the current study (Fig 4), supporting the importance of evaluating the F–v profiles of both SJ and CMJ.

## Conclusions

Results showed that the reactive strength was not associated with any vertical F–v profile parameter, whereas the maximum strength was associated with the theoretical maximum force (i.e., F0) of SJ and CMJ, and also with slope of the F–v relationship (i.e., SFv) of SJ and theoretical maximum power (i.e., Pmax) of CMJ. These results indicate that maximum strength is a more important indicator than reactive strength in improving vertical F–v profiles. Furthermore, the importance of maximum strength can vary depending on whether the practitioner wants to maximize the performance of SJ or CMJ.

## Supporting information

**S1 Data. Original data.**
(XLSX)

## Acknowledgments

The authors would like to thank all participants for their cooperation and Enago (www.enago. jp) for the English language review.

## Author Contributions

**Conceptualization:** Takuya Nishioka.

**Data curation:** Takuya Nishioka.

**Formal analysis:** Takuya Nishioka.

**Investigation:** Takuya Nishioka.

**Methodology:** Takuya Nishioka.

**Supervision:** Junichi Okada.

**Writing – original draft:** Takuya Nishioka.

**Writing – review & editing:** Takuya Nishioka, Junichi Okada.

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
