## [Decision Letter · Decision Letter 0]

7 Sep 2022

PONE-D-22-17503Associations of maximum and reactive strength indicators with force–velocity profiles obtained from squat jump and countermovement jumpPLOS ONE

Dear Dr. Nishioka,

Thank you for submitting your manuscript to PLOS ONE. After careful consideration, we feel that it has merit but does not fully meet PLOS ONE’s publication criteria as it currently stands. Therefore, we invite you to submit a revised version of the manuscript that addresses the points raised during the review process.

We look forward to receiving your revised manuscript.

Kind regards,

Javier Abián-Vicén, Ph.D.

Academic Editor

PLOS ONE

Journal Requirements:

2. You indicated that you had ethical approval for your study. If minors were involved, in your Methods section, please ensure you have also stated whether you obtained consent from parents or guardians of the minors included in the study or whether the research ethics committee or IRB specifically waived the need for their consent.

3. Please indicate the place (location) and time (dates) when the study was conducted.

Reviewers' comments:

Reviewer's Responses to Questions

**Comments to the Author**

1. Is the manuscript technically sound, and do the data support the conclusions?

Reviewer #1: Partly

Reviewer #2: Partly

2. Has the statistical analysis been performed appropriately and rigorously? 

Reviewer #1: Yes

Reviewer #2: Yes

3. Have the authors made all data underlying the findings in their manuscript fully available?

Reviewer #1: Yes

Reviewer #2: Yes

4. Is the manuscript presented in an intelligible fashion and written in standard English?

Reviewer #1: Yes

Reviewer #2: Yes

5. Review Comments to the Author

Reviewer #1: General Comments

The authors have sought to investigate the strength of relationship between a measure of maximal lower limb strength, and indicator of reactive strength quality and descriptive metrics of the force-velocity relationship obtained from squat jumps and countermovement jumps. The investigation has merit however I am concerned by the underlying assumption that the F-V relationship is linear when in fact it is curvilinear with an exponential decay from a maximum force at zero velocity down to zero force with a theoretical maximal velocity. Coupled with this issue is the fact that the authors have only tested each movement using two loads, an unloaded (bodyweight condition) and another with an external load of 40% 1RM with no justification for this loading condition or pattern. Interestingly the authors did perform a warm up across 4 different loading conditions but only at an 80% effort. The manuscript has been well written and it is clear that the authors have used an English language service.

Specific Comment

Ln14; Please change 'abilities' to 'properties'

Ln21; The section "...and drop jump (DJ) reactive strength index (RSI)." is confusing as it from my reading perspective it appears as if the authors calculated F-V profile information from the DJ as well which is not the case. Please re-word to clarify.

Ln37; Please change 'abilities' to 'properties'

Ln58;Please change 'abilities' to 'properties'

Ln65-67; This sentence is not a hypothesis and should be re-worded appropriately

Ln67; I have no issues with authors reanalysing published data to answer a new question, however this sentence is clumsily worded and should be re-worded.

Ln72; Change the word 'for' for 'across'

Ln74-77 and 188-193; This is a problematic flaw in the design of the investigation as measuring F-V properties from only two loads is not a comprehensive profile and negates the ability to determine whether the relationship is linear or not. The classical F-V relationship from isolated muscle fibres is a curvilinear relationship with an exponential decay. As such in multi-point movements, while the multifaceted characteristics of the muscle tendon unit may mask some of this curvilinear relationship a linear relationship should not be assumed. How certain are the authors that the relationship is linear? The authors need to justify relationship modelling choice, and the use of only 2 testing loads especially considering that the warm up protocol included 4 different loads.

Ln254; The discussion is likely to require a substantial re-write if changes are made to the choice of relationship model. The authors could also choose to report the relationship from actually recorded values rather than the calculated theoretical values which rely on the choice of model. Importantly it is these recorded values that any specifically undertaken training would change.

Figure 1, 2, 3 and 4; Please include the 95% CI for the line of best fit/regression when included

Reviewer #2: Congratulations for your work. Some concerns should be resolve before publication.

The short title doesn't talk about jump performance.

In your introduction you should talk about the term that you want to resolve with your research. Please rewrite this section giving more information about the F-v measurement importance.

In the first appearance should be stretch-shorting cycle (SSC). Review all the abbreviations throughout the manuscript and remember that abstract section is independent.

Line 78 What was the reason to have this range of hours?

Line 88: Delete mean ± SD

Line 107: rest (60-90 s)

Line 115: How and when did you estimate the individual RM?

Line 116: At the end of the warm up you were close to RM, it was this weigth used at the initial attemps?

Line 136: Why absolute value and not RM percentage?

Line 143: under the same environmental conditions

Line 203: What was the result of normality test?

Line 259: Explain here your hypothesis in relation with your results.

6. PLOS authors have the option to publish the peer review history of their article (what does this mean?). If published, this will include your full peer review and any attached files.

Reviewer #1: **Yes: **Dale Wilson Chapman

Reviewer #2: No

---

## [Author Response · Author response to Decision Letter 0]

20 Sep 2022

Reviewer #1: General Comments

The authors have sought to investigate the strength of relationship between a measure of maximal lower limb strength, and indicator of reactive strength quality and descriptive metrics of the force-velocity relationship obtained from squat jumps and countermovement jumps. The investigation has merit however I am concerned by the underlying assumption that the F-V relationship is linear when in fact it is curvilinear with an exponential decay from a maximum force at zero velocity down to zero force with a theoretical maximal velocity. Coupled with this issue is the fact that the authors have only tested each movement using two loads, an unloaded (bodyweight condition) and another with an external load of 40% 1RM with no justification for this loading condition or pattern. Interestingly the authors did perform a warm up across 4 different loading conditions but only at an 80% effort. The manuscript has been well written and it is clear that the authors have used an English language service.

　Response: Previous studies (1) have shown that the force–velocity profiles of both SJ and CMJ demonstrate a linear relationship. Although these force–velocity profiles are commonly measured using 5–6 different loads (2), García-Ramos et al. (2) reported that accurate measurements recorded using two distant loads (i.e., two-point method). In the present study, to eliminate the effects of interday variability and minimize participants’ fatigue, the force–velocity profiles of SJ and CMJ were measured on the same day using a two-point method. In addition, 40% 1RM was similar to the maximum load commonly used for force–velocity profiling (3) and was considered appropriate for the two-point method. Furthermore, this is reflected by the high reliability of this experiment (see Table 1). Additionally, we have added the following sentence explaining the linear force-velocity relationship for vertical jump:

 “Owing to the fact that the force–velocity relationship of vertical jumps is linear [7,8],” (Lines 203–204)

1. Jiménez-Reyes P, Samozino P, Cuadrado-Peñafiel V, Conceição F, González-Badillo JJ, Morin JB. Effect of countermovement on power-force-velocity profile. Eur J Appl Physiol. 2014;114: 2281-2288.doi: 10.1007/s00421-014-2947-1 

2. García-Ramos A, Pérez-Castilla A, Jaric S. Optimisation of applied loads when using the two-point method for assessing the force-velocity relationship during vertical jumps. Sports Biomech. 2021;20: 274-289.doi: 10.1080/14763141.2018.1545044 

3. Rivière JR, Rossi J, Jimenez-Reyes P, Morin JB, Samozino P. Where does the one-repetition maximum exist on the force-velocity relationship in squat? Int J Sports Med. 2017;38: 1035-1043.doi: 10.1055/s-0043-116670 

Specific Comment

Ln14; Please change 'abilities' to 'properties'

　Response: We have incorporated this change as per your suggestion.

Ln21; The section "...and drop jump (DJ) reactive strength index (RSI)." is confusing as it from my reading perspective it appears as if the authors calculated F-V profile information from the DJ as well which is not the case. Please re-word to clarify.

　Response: Based on your suggestion, we revised the sentence as follows:

 “On the second day, jump performances were measured to calculate the drop jump (DJ) reactive strength index (RSI), and the parameters of F–v profiles (theoretical maximum force [F0], velocity [V0], power [Pmax], and slope of the linear F–v relationship [SFv]) obtained from SJ and CMJ.” (Lines 18–22)

Ln37; Please change 'abilities' to 'properties'

　Response: We have incorporated this change as per your suggestion.

Ln58;Please change 'abilities' to 'properties'

　Response: We have incorporated this change as per your suggestion.

Ln65-67; This sentence is not a hypothesis and should be re-worded appropriately

　Response: As you pointed out, the wording was somewhat ambiguous. Therefore, we revised the sentence as follows:

 “We hypothesized that the maximum and reactive strength indicators would be positively associated with the vertical F–v profiles (i.e., F0, V0, and Pmax).” (Lines 76–77)

Ln67; I have no issues with authors reanalysing published data to answer a new question, however this sentence is clumsily worded and should be re-worded.

　Response: Based on your suggestion, we revised the following sentence:

 “In the present study, we analyzed data from a previous study [21] for a completely different purpose.” (Lines 77–78)

Ln72; Change the word 'for' for 'across'

　Response: We have incorporated this change as per your suggestion.

Ln74-77 and 188-193; This is a problematic flaw in the design of the investigation as measuring F-V properties from only two loads is not a comprehensive profile and negates the ability to determine whether the relationship is linear or not. The classical F-V relationship from isolated muscle fibres is a curvilinear relationship with an exponential decay. As such in multi-point movements, while the multifaceted characteristics of the muscle tendon unit may mask some of this curvilinear relationship a linear relationship should not be assumed. How certain are the authors that the relationship is linear? The authors need to justify relationship modelling choice, and the use of only 2 testing loads especially considering that the warm up protocol included 4 different loads.

　Response: As you mentioned, the force–velocity relationship of muscle fibers is curvilinear (1). However, as mentioned above, the force–velocity profiles of SJ and CMJ measured in this study have a linear relationship (2). This difference is explained by the fact that segmental dynamics in multi-joint movements reduce external forces at high velocities (3). In addition, a higher number of measurement trials may lead to subjects’ fatigue and inaccurate assessment of performance (4). Based on the above, we measured the force–velocity profiles of vertical jumps using the two-point method (4). 

1. Ritchie JM, Wilkie DR. The dynamics of muscular contraction. J Physiol. 1958;143: 104-113.doi: 10.1113/jphysiol.1958.sp006047 

2. Jiménez-Reyes P, Samozino P, Cuadrado-Peñafiel V, Conceição F, González-Badillo JJ, Morin JB. Effect of countermovement on power-force-velocity profile. Eur J Appl Physiol. 2014;114: 2281-2288.doi: 10.1007/s00421-014-2947-1 

3. Bobbert MF. Why is the force-velocity relationship in leg press tasks quasi-linear rather than hyperbolic? J Appl Physiol (1985). 2012;112: 1975-1983.doi: 10.1152/japplphysiol.00787.2011 

4. García-Ramos A, Pérez-Castilla A, Jaric S. Optimisation of applied loads when using the two-point method for assessing the force-velocity relationship during vertical jumps. Sports Biomech. 2021;20: 274-289. doi: 10.1080/14763141.2018.1545044 

Ln254; The discussion is likely to require a substantial re-write if changes are made to the choice of relationship model. The authors could also choose to report the relationship from actually recorded values rather than the calculated theoretical values which rely on the choice of model. Importantly it is these recorded values that any specifically undertaken training would change.

　Response: As noted above, it is reasonable to analyze the force–velocity profiles of SJ and CMJ via linear regression. However, even if changes in recorded values (e.g., jump height) are important, Samozino et al. (1) have reported that theoretical values such as F0, V0, and Pmax directly contribute to improvements in recorded performance (i.e., unloaded jump height). This fact has been further confirmed through training intervention studies (2,3). Therefore, it seems reasonable to calculate theoretical values from the force–velocity relationship of vertical jumps by linear regression and discuss those variables as important performance. The validity of our argument is supported by a recent systematic review on force–velocity profiles (4).

1. Samozino P, Rejc E, Di Prampero PE, Belli A, Morin JB. Optimal force-velocity profile in ballistic movements–altius: citius or fortius? Med Sci Sports Exerc. 2012;44: 313-322. doi: 10.1249/MSS.0b013e31822d757a PubMed: 21775909

2. Jiménez-Reyes P, Samozino P, Brughelli M, Morin JB. Effectiveness of an individualized training based on force-velocity profiling during jumping. Front Physiol. 2016;7: 677. doi: 10.3389/fphys.2016.00677 PubMed: 28119624 

3. Jiménez-Reyes P, Samozino P, Morin JB. Optimized training for jumping performance using the force-velocity imbalance: individual adaptation kinetics. PLOS ONE. 2019;14: e0216681. doi: 10.1371/journal.pone.0216681 PubMed: 31091259

4. Baena-Raya A, García-Mateo P, García-Ramos A, Rodríguez-Pérez MA, Soriano-Maldonado A. Delineating the potential of the vertical and horizontal force-velocity profile for optimizing sport performance: A systematic review. J Sports Sci. 2022;40: 331-344. doi: 10.1080/02640414.2021.1993641 

Figure 1, 2, 3 and 4; Please include the 95% CI for the line of best fit/regression when included

　Response: We incorporated this information as per your suggestion.

Reviewer #2: Congratulations for your work. Some concerns should be resolve before publication.

The short title doesn't talk about jump performance.

　Response: We changed the short title from “Strength indicators and force–velocity profiles” to “Strength indicators and vertical force–velocity profiles.” “Vertical force–velocity profiles” indicates the force–velocity profiles obtained from jump performance (1).

 1. Morin JB, Samozino P. Interpreting power-force-velocity profiles for individualized and specific training. Int J Sports Physiol Perform. 2016;11: 267-272. doi: 10.1123/ijspp.2015-0638 PubMed: 26694658 

In your introduction you should talk about the term that you want to resolve with your research. Please rewrite this section giving more information about the F-v measurement importance.

　Response: We added the follow sentence as per your suggestion:

 “F0 represents the maximal concentric force output (per body mass unit) that an athlete’s lower limbs can theoretically produce during a ballistic push-off, and V0 represents the maximal extension velocity of the athlete’s lower limbs during a ballistic push-off [7]. Both parameters are extrapolated from the linear F–v relationship in loaded jump squats, but not from peak values during a single jump [7]. Pmax, computed as Pmax = F0∙V0/4, represents the maximal power output capability of the athlete’s lower limb neuromuscular system (per body mass unit) in concentric and ballistic extension motions, and SFv is an index of the athlete’s individual balance between force and velocity capabilities, in which there is an optimal balance for a given individual to maximize the jump performance [7].” (Lines 43–52)

In the first appearance should be stretch-shorting cycle (SSC). Review all the abbreviations throughout the manuscript and remember that abstract section is independent.

　Response: We carefully checked the abbreviations, including the abstract section and found no further issues.

Line 78 What was the reason to have this range of hours?

　Response: After high-intensity resistance exercise, full recovery of neuromuscular performance takes 72 h (1). In addition, since an excessively long interval between the preliminary and main measurements could alter an individual’s maximal strength, we chose 72–144 h.

 1. Thomas K, Brownstein CG, Dent J, Parker P, Goodall S, Howatson G. Neuromuscular fatigue and recovery after heavy resistance, jump, and sprint training. Med Sci Sports Exerc. 2018;50: 2526-2535. doi: 10.1249/MSS.0000000000001733 

Line 88: Delete mean ± SD

　Response: We deleted that expression as per your suggestion.

Line 107: rest (60-90 s)

　Response: We edited the text accordingly.

Line 115: How and when did you estimate the individual RM?

　Response: As all participants in this study had previous resistance training experience, they knew their own approximate half squat 1RM. In addition, before measuring an individual’s true 1RM, we had to set the load based on the subject’s own known approximate 1RM (estimated 1RM). Therefore, the estimated 1RM was used as the basis for setting the load during the warm-up for the 1RM measurement. This method has been used in a previous study (1).

 1. Cormie P, McGuigan MR, Newton RU. Changes in the eccentric phase contribute to improved stretch-shorten cycle performance after training. Med Sci Sports Exerc. 2010;42: 1731-1744. doi: 10.1249/MSS.0b013e3181d392e8 

Line 116: At the end of the warm up you were close to RM, it was this weigth used at the initial attemps?

　Response: The 90% estimated 1RM corresponded to the load at the end of the warm-up, but not the initial 1RM attempt. In the manuscript, we indicated that the maximal lift attempt was performed after warm-up. 

Line 136: Why absolute value and not RM percentage?

　Response: Since relative loads were unlikely to differ among participants at light loads, only 20 kg was set as the absolute load. In addition, this simplified the experimental procedure.

Line 143: under the same environmental conditions

　Response: We incorporated this change as per your suggestion.

Line 203: What was the result of normality test?

　Response: Please review the nonparametric data shown in the supplemental file (“original data”). 

Line 259: Explain here your hypothesis in relation with your results.

　Response: We added the following sentence as per your suggestion:

 “We hypothesized that the maximum and reactive strength indicators would be positively associated with the vertical F–v profiles (i.e., F0, V0, and Pmax).” (Lines 277–279)

---

## [Decision Letter · Decision Letter 1]

12 Oct 2022

Associations of maximum and reactive strength indicators with force–velocity profiles obtained from squat jump and countermovement jump

PONE-D-22-17503R1

Dear Dr. Nishioka,

We’re pleased to inform you that your manuscript has been judged scientifically suitable for publication and will be formally accepted for publication once it meets all outstanding technical requirements.

Kind regards,

Javier Abián-Vicén, Ph.D.

Academic Editor

PLOS ONE

Additional Editor Comments (optional):

I'm going to accept the paper but the authors must correct the errors identified by reviewer 1 in figures 1, 2 and 4.

Fig 1; Check the SJ V0 and SJ Pmax panels as there are two p values

Fig 2; Check the SJ V0 and SJ Pmax panels as there are two p values

Fig 4; There are two p values provided in the V0 and Max panels?

Congratulations for your paper!

Reviewers' comments:

Reviewer's Responses to Questions

**Comments to the Author**

1. If the authors have adequately addressed your comments raised in a previous round of review and you feel that this manuscript is now acceptable for publication, you may indicate that here to bypass the “Comments to the Author” section, enter your conflict of interest statement in the “Confidential to Editor” section, and submit your "Accept" recommendation.

Reviewer #1: (No Response)

Reviewer #2: All comments have been addressed

2. Is the manuscript technically sound, and do the data support the conclusions?

Reviewer #1: Yes

Reviewer #2: Yes

3. Has the statistical analysis been performed appropriately and rigorously? 

Reviewer #1: Yes

Reviewer #2: Yes

4. Have the authors made all data underlying the findings in their manuscript fully available?

Reviewer #1: Yes

Reviewer #2: Yes

5. Is the manuscript presented in an intelligible fashion and written in standard English?

Reviewer #1: Yes

Reviewer #2: Yes

6. Review Comments to the Author

Reviewer #1: I thank the authors for their comprehensive response to the initial review.

Fig 1; Check the SJ V0 and SJ Pmax panels as there are two p values

Fig 2; Check the SJ V0 and SJ Pmax panels as there are two p values

Fig 4; There are two p values provided in the V0 and Max panels?

Reviewer #2: Congratulations for your work agian. The manuscript has been improved and now has the quality to be published in PloSOne.

7. PLOS authors have the option to publish the peer review history of their article (what does this mean?). If published, this will include your full peer review and any attached files.

Reviewer #1: **Yes: **Dale Wilson Chapman

Reviewer #2: No

---

## [Editor Report · Acceptance letter]

14 Oct 2022

PONE-D-22-17503R1 

Associations of maximum and reactive strength indicators with force–velocity profiles obtained from squat jump and countermovement jump 

Dear Dr. Nishioka:

I'm pleased to inform you that your manuscript has been deemed suitable for publication in PLOS ONE. Congratulations! Your manuscript is now with our production department. 

Kind regards, 

on behalf of

Dr. Javier Abián-Vicén 

Academic Editor

PLOS ONE